# Preparation of ECTFE Porous Membrane for Dehumidification of Gaseous Streams through Membrane Condenser

**DOI:** 10.3390/membranes12010065

**Published:** 2022-01-01

**Authors:** Jun Pan, Kun Chen, Zhaoliang Cui, Omar Bamaga, Mohammed Albeirutty, Abdulmohsen Omar Alsaiari, Francesca Macedonio, Enrico Drioli

**Affiliations:** 1State Key Laboratory of Materials-Oriented Chemical Engineering, College of Chemical Engineering, Nanjing Tech University, Nanjing 211816, China; paju152@163.com (J.P.); kunchen0501@163.com (K.C.); 2National Engineering Research Center for Special Separation Membrane, Nanjing Tech University, Nanjing 211816, China; 3Jiangsu National Synergetic Innovation Center for Advanced Materials (SICAM), Nanjing Tech University, Nanjing 211816, China; 4Center of Excellence in Desalination Technology, King Abdulaziz University, P.O. Box 80200, Jeddah 21589, Saudi Arabia; mbeirutty@kau.edu.sa (M.A.); aoalsaiari@kau.edu.sa (A.O.A.); 5Mechanical Engineering Department, King Abdulaziz University, P.O. Box 80200, Jeddah 21589, Saudi Arabia; 6Research Institute on Membrane Technology, ITM-CNR, Via Pietro Bucci 17/C, 87036 Rende, Italy; f.macedonio@itm.cnr.it (F.M.); e.drioli@itm.cnr.it (E.D.)

**Keywords:** membrane condenser, hydrophobic membrane, ECTFE membrane, thermally induced phase separation

## Abstract

Due to the good hydrophobicity and chemical resistance of poly(ethylene trifluoroethylene) (ECTFE), it has been an attractive potential material for microfiltration, membrane distillation and more. However, few porous hydrophobic ECTFE membranes were prepared by thermally induced phase separation (TIPS) for membrane condenser applications. In this work, the diluent, di-n-octyl phthalate (DnOP), was selected to prepare the dope solutions. The calculated Hassen solubility parameter indicated that ECTFE has good compatibility with DnOP. The corresponding thermodynamic phase diagram was established, and it has been mutually verified with the bi-continuous structure observed in the SEM images. At 30 wt% ECTFE, the surface contact angle and liquid entry pressure reach their maximum values of 139.5° and 0.71 MPa, respectively. In addition, some other basic membrane properties, such as pore size, porosity, and mechanical properties, were determined. Finally, the prepared ECTFE membranes were tested using a homemade membrane condenser setup. When the polymer content is 30 wt%, the corresponding results are better; the water recovery and condensed water yield is 17.6% and 1.86 kg m^−2^ h^−1^, respectively.

## 1. Introduction

Membrane contactors are membrane systems designed to achieve contact between two phases. Examples of membrane contactors include membrane distillation (MD), membrane crystallization (MCr), and membrane emulsification. These systems have been used in wastewater treatment, desalination, the medical industry, and other applications [1,2]. Along with the development of society and technology, many new problems occur and may threaten the environment. The problem of waste flue gases, which come from the power plant, steel industry, coking industry, and other industrial facilities, represents one of the challenges to be solved. Many toxic substances, such as nitrogen oxides, sulfur dioxide, smoke, and other harmful things, are produced in the fabrication process and are ejected with the flue gases. The wet removal methods are the most common methods utilized to remove toxic gases to meet the emission standards. However, in these methods, the humidity of treated gas increases and even reaches near saturation levels. When the treated and wetted gaseous stream enters the atmosphere, the vapor in the stream can reach the supersaturated state since the hotter stream can be cooled as soon as it comes in contact with the surrounding colder air. The water vapor condenses to tiny droplets and is atomized to form smoke, the so-called ‘white smoke’ phenomenon. This always tends to cause air pollution and light pollution. When these tiny droplets lose moisture, they leave countless small, suspended solids in the atmosphere, i.e., aerosols, which is also the main reason for smog formation. In addition, some harmful substances remaining in the high humidity steam are also the reason for acid rain. Most importantly, the direct emission of moisture with such a high-water content causes a waste of valuable water resources and dramatically increases the production cost of the industries.

In response to this problem, the membrane condenser based on hydrophobic membranes was first introduced by Drioli et al. (2013) [3], the principle of which is shown in Figure 1. This is a relatively new process of the membrane contactors’ group, designed initially to dehumidify the white, high moisture waste stream to recover the water vapor. Due to the hydrophobic nature of the applied membranes, the vapor in the feed gas starts to condense when it comes in contact with the membrane surface because of the temperature difference between the gas and the membrane. Finally, the condensed water is rolled down the membrane surface and collected on the upstream side while the dehumidified dry air penetrates through the membrane to the downstream side.

Though some other concepts for using membrane contactors for moisture condensation were suggested earlier, the hydrophobic membrane condenser operation is still an innovative step with many irreplaceable advantages. Traditional technologies, such as cooling with condensation, liquid and solid sorption, and cryogenic separation [4], provide simple solutions to recover water. The main limitations of these technologies are corrosion, ash fouling, and extra physical and chemical costs. However, the membrane technology is environmentally friendly, highly efficient, and does not cause secondary pollution during the treatment process. Cao et al. [5] compared three membrane-based dehydration models, including dense, hydrophilic, and hydrophobic membranes. For the first one, extra-high pressure is required to force the condensed water to penetrate through the membrane. Though the quality of the recovered water can be kept well, it is bound to increase energy consumption [6]. In the hydrophilic membrane model, also called the transport membrane condenser, the water vapor condenses inside the membrane pores before it passes to the downstream side of the membrane. Hence, the quality of the recovered water can be affected by the pollutants contained in the flue gas [7]. The innovative membrane condenser concept ensures the high quality of the recovered water and simultaneously enables heat and mass transfer at lower energy consumption.

To date, the fluorinated polymer family is the most popular for preparing hydrophobic membranes, including polyvinylidene fluoride (PVDF) and its copolymer (PVDF-HFP), (PVDF-CTFE), (PVDF-TrFE), and polytetrafluoroethylene (PTFE) [8,9,10]. Indeed, some other non-fluorinated polymers, such as polypropylene (PP), or other novel materials, such as graphene oxide (GO) and carbon nanotubes (CNTs), are also used to prepare hydrophobic membranes [11,12,13,14,15]. As for ethylene-chlorotrifluoroethylene copolymer (ECTFE), it can be considered a relatively new member of this fluoropolymer family. ECTFE is a semi-crystalline, thermoplastic fluorinated copolymer polymerized by ethylene and chlorotrifluoroethylene at a molar ratio of one to one. As for the ECTFE, the fluorine content can be up to about 40%, which contributes to its good hydrophobicity and is even better than PVDF. There are some studies about the MD application of ECTFE membranes indicating good anti-wetting for desalination [16,17,18,19,20,21,22,23,24]. Moreover, ECTFE has excellent chemical corrosion resistance against acid, alkali, oxidant, and reducing and corrosive agents. It has a significant advantage over the non-alkali-resistant PVDF, though the membrane formation property of the latter is easier. Therefore, the ECTFE is also used as an anti-corrosion coating layer of pipes, tubes, and other equipment [25,26]. Moreover, the mechanical properties, abrasion resistance, and creep resistance of ECTFE are still excellent [27,28]. Therefore, ECTFE is a potential candidate material for preparing hydrophobic microporous membranes for membrane condensers, mainly when the membrane condenser is used to treat wasted streams containing acid, ammonia, etc. There is no doubt that the PTFE still has the best hydrophobicity among fluoropolymers, but the relatively complex fabrication process and expensive cost make it out of our consideration [10].

However, owing to its excellent chemical resistance, ECTFE is not soluble in the typical solvents at room or low temperatures. Therefore, ECTFE membranes are usually prepared by thermally induced phase separation (TIPS) at a relatively high temperature. Table 1 summarizes the published work on ECTFE membrane preparation. The table indicates that there is only a little work on ECTFE membranes compared with PVDF or other membranes, and almost all researchers have used the TIPS method to prepare ECTFE membranes. In the TIPS method, the polymer is dissolved at a high temperature, and the phase separation happens at a lower or colder temperature. Usually, the membranes prepared by TIPS have a narrower pore size distribution and higher mechanical properties than those prepared by non-solvent-induced phase separation [29,30].

In this work, the TIPS method is used to prepare ECTFE hydrophobic flat sheet membranes using DnOP. The compatibility of ECTFE and DnOP is calculated, and the phase diagram of this ECTFE/DnOP system is drawn. One ECTFE concentration gradient has been prepared. The prepared membranes were characterized and analyzed concerning morphology, contact angle, liquid entry pressure, pore size, porosity, and mechanical properties. Finally, the prepared ECTFE membranes were tested under the membrane condenser process to determine the dehumidification performance, which can be analyzed from the water recovery and the condensate flow. The study involved in this work proposes a potential preparation method and application of ECTFE membranes.

## 2. Materials and Methods

### 2.1. Materials

ECTFE was supplied by Zhejiang Chemical Research Institute (Hangzhou, China). DnOP and kerosene were purchased from Aladdin Biochemical Technology Co., Ltd. (Shanghai, China). Ethanol was purchased from Sinopharm Chemical Reagent Co., Ltd. (Shanghai, China). GQ-16, which acts as the wetting agent for pore size measurement, was purchased from Jiangsu Gaoqian Function Material Co., Ltd. (Nanjing, China). Table 2 shows the basic properties of ECTFE and DnOP.

### 2.2. Membrane Preparation

ECTFE membranes were prepared by thermally induced phase separation. Firstly, the ECTFE powder was dried at 40 °C overnight. Then the desired amount of ECTFE and DnOP were weighed and placed in one three-necked flask. Five different mixtures of ECTFE and DnOP were prepared where the range of ECTFE content in the mixture was varied from 15 to 35 wt%. The mixture was stirred in an oil bath at 250 °C for three hours until a homogeneous casting solution was formed, followed by one hour of defoaming. Then, the appropriate amount of casting solution was poured into the pre-heated customized stainless steel membrane mold. Then, the mold was pressurized for 15 min to allow the polymer solution to cast and shape. Finally, the mold was taken out and quenched into a water bath at 25 °C to induce phase separation. The residual diluent contained in the ECTFE membrane was extracted by ethanol overnight. After cleaning with pure water, the ECTFE membranes were obtained by freeze-drying.

### 2.3. Characterization

#### 2.3.1. Basic Properties of ECTFE Membrane

Membrane morphology and topography were observed using a field-emission scanning electron microscope (FESEM, Hitachi S4800, Tokyo, Japan) and an atomic force microscope (AFM, Bruke Icon, Karlsruhe, Germany), respectively. For such membrane condenser processes based on a hydrophobic porous membrane, the water repulsion properties are fundamental, including water contact angle (WCA) and liquid entry pressure (LEP) measurement by a commercial contact angle instrument (Dataphysics OCA 25, Filderstadt, Germany) and a homemade dead-end filtration setup, respectively. The relationship between WCA, LEP, and pore size can be analyzed using Equation (1). Mechanical properties, including tensile strength and elongation at break, were measured by a tensile strength testing instrument (Model SH-20, Wenzhou Shandu Instrument Co., Wenzhou, China).
(1)$$LEP_{w}=\frac{-2B\times \gamma_{L}\times \cos\theta }{r_{max}}$$
where *B* is the dimensionless geometric parameter, which can be equal to 1 for the assumed perfectly cylindrical pores, $\gamma_{L}$ is the liquid surface tension, θ is the CA in degrees, and $r_{max}$ is the maximum pore size.

These measurements were conducted three times for each sample, the procedures of which can be found in our previous study [16,17].

#### 2.3.2. Phase Diagram

For the TIPS method, the phase diagram is a necessary approach to analyze the phase separation behavior and forecast the possible membrane structure. The Hansen Solubility Parameter Theory (HSP) is considered an effective method to describe the interaction between the polymer and diluent, presented by the HSP distance (*R*), Equation (2) presents [42]: (2)$$R^{2}=4{\left(\delta_{d,p}-\delta_{d,d}\right)}^{2}+{\left(\delta_{p,p}-\delta_{p,d}\right)}^{2}+{\left(\delta_{h,p}-\delta_{h,d}\right)}^{2}$$
where $\delta_{d}$ is the dispersion force, $\delta_{p}$ is the polar force, and $\delta_{h}$ is the hydrogen bond force, respectively (Note: $\delta_{A,B}$, *A* is the type of force, *B* is the polymer or diluent). 

The phase diagram consists of two curves: the crystallization temperature curve and the cloud point curve. The crystallization temperature is tested by a differential scanning calorimetry (DSC, Q-20, Newcastle City, DE, USA), and the cloud point temperature is tested by a polarizing microscope (XPV-800E, Shanghai, China). The detailed measurement procedures have been displayed in our previous study [29].

#### 2.3.3. Membrane Condenser

Figure 2 illustrates the homemade membrane condenser experimental device, and Table 3 indicates the operation parameters of the membrane condenser tests. The mixed water vapor and the air were used to simulate the high humidity exhaust gas. First, the humidification and buffer tanks were placed inside a water bath and then heated and kept at the operating temperature. Then, the dry gas flow from an air compressor went into the humidification tank to increase the moisture content. The existence of the buffer tank was for the complete mixing of air and vapor to achieve a saturation state. Temperature and humidity meters were mounted at the outlet of the buffer tank to measure the temperature and humidity values. When the heated saturated humid gas enters the membrane module, the heat exchange happens on the membrane surface because of the temperature difference between the gas and the membrane. Then, the vapor molecules started to condensate on the membrane surface. Due to the hydrophobic nature of the membrane, the condensed water droplets cannot penetrate through the membrane pores. Instead, the droplets rolled down from the membrane surface. At the same time, the non-condensable gas was discharged through the membrane pores. For the detailed mass balance calculation, it has been reported in study [3]. In addition, it is necessary to check whether the used membrane module is leaking before putting it into the constant temperature oven. Moreover, it should be noted that all containers and flow tubes were wrapped with insulation cotton to avoid unnecessary heat loss. In addition, the condensed water was collected every 30 min.

The membrane condenser operation is applied for the dehumidification of wet gas streams. The condensate flow (*J*, kg‧m^−2^‧h^−1^) and water recovery (*R*, wt%) are used to evaluate the dehumidification performance of the process, which are calculated by the following equations:(3)$$R=\frac{\Delta m}{M}$$
(4)$$J=\frac{\Delta m}{A\Delta t}$$
where Δm (kg) is the weight of liquid water on the retentate side, *M* (kg) is the total weight of water vapor contained in the stream during the same operating period, Δt (h) is the interval operation time, and *A* (m^2^) is the effective membrane area.

## 3. Results and Discussions

### 3.1. Phase Diagram

The choice of the diluent to prepare ECTFE membranes by TIPS is an extremely significant step. By the calculation of Equation (3), the obtained *R*-values of ECTFE and other diluents are shown in Table 4. The smaller the *R*-value, the stronger the compatibility between the ECTFE and diluent is. From Table 4, the *R*-value of the ECTFE/DnOP system is 5.97, lower than most of the related published work [16,17,43], indicating that the compatibility is good and hopeful.

Figure 3 is the thermodynamic phase diagram of the ECTFE/DnOP binary system, from which the compatibility of the polymer/diluent can be proven and possible phase separation behavior can be analyzed. Certainly, the membrane structure can also be forecasted from it. It should be noted that the viscosity of the dope solution is too high when the ECTFE content exceeds 40 wt%. Therefore, the range of the polymer content varies from 15 to 35 wt%, as shown in Figure 3.

From Figure 3, it can be seen that the crystallization temperature seems to be proportional to the polymer concentration. However, there is a small difference for cloud point; with the increasing of the ECTFE content, the cloud point temperature firstly increased and then decreased. Figure 3 shows that the cloud point temperature and the crystallization temperature curves would have one intersection point. It is generally called the monotectic point [42]; the corresponding ECTFE content and temperature is around 40 wt% and 185 °C, respectively. Moreover, within the tested polymer content, the cloud point curve is always higher than the crystallization temperature curve, which means a liquid–liquid (L–L) phase separation zone exists in the ECTFE/DnOP system. When the stable doping solution cools down along Path 1 until it reaches the cloud point curve, the L–L phase separation happens first. The two phases, polymer-rich and polymer-poor phases appear in the system. The former becomes the membrane matrix, and the latter is the membrane pores. Usually, when the polymer/diluent experience the same routine as Path 1, the bi-continuous structure would eventually form [44]. Then for a higher polymer content, over 40 wt%, the liquid–solid (L–S) phase separation would happen firstly during the cooling rate for Path 2. Since the crystallization temperature is higher than the corresponding cloud point temperature, there would only be L–S phase separation happening. The rich polymer phase solidifies and crystallizes directly, and the poor phase exists in the polymer lamellae. Finally, the porous and spherulitic structure can be obtained [45].

In the end, in this work, there is always an L–L phase separation zone for all of the tested polymer contents. Therefore, it can be inferred that the prepared ECTFE membranes would present a bi-continuous structure. 

### 3.2. Membrane Characterization

#### 3.2.1. Morphology and Topography of ECTFE Membrane

Table 5 shows the SEM and AFM images of the prepared membranes, from which the surface and cross-section structure can be seen clearly. According to the SEM images, all of the prepared membranes presented a porous structure. The whole structure gets denser when the polymer content increases. It can be inferred that the prepared membranes have bi-continuous structures from the cross-section images. This is consistent with the assumption and analysis in Section 3.1. It should be noted that the SEM images of the membrane surface structure show many raised ridges on the membrane surface after the preparation process. These ridges are not observed in other published work [18,31,37,38,40]. The ridges increase the membrane surface roughness, reducing the adhesion and promoting heterogeneous condensation of water vapor on the membrane surface during the membrane condenser operating period. Certainly, these 3D AFM images can also reflect the rough ups and downs of the topography.

#### 3.2.2. Hydrophobicity Measurement of ECTFE Membrane

In this work, the hydrophobic membrane acts as the core part of the membrane condenser operation. The vapor condenses and forms droplets on the membrane surface and is rejected due to the hydrophobic nature of the membrane. There is a lower possibility of membrane wetting when the membrane has a high hydrophobicity. High hydrophobicity is required for a longer service membrane life and a more stable membrane condenser process. In addition, the liquid entry pressure (LEP) value indicates the maximum pressure to maintain the process free of wetting, which can also be used to prove the anti-wetting and hydrophobic properties. In Figure 4a,b, the contact angle (CA) and liquid entry pressure of water (LEP_w_) of the prepared membranes are shown, respectively.

In Figure 4a, the CA values of all of the prepared ECTFE membranes exceed 120°; the value first increases and then decreases along with the polymer content. When the ECTFE content becomes 30 wt%, the CA value reaches the maximum, 139.5°. The numerous C–F bonds that exist in ECTFE chains contribute to this good result. In addition to the polymer properties, the surface CA is also related to the membrane surface roughness, as shown in Table 6. The maximum (*R*_max_) and average surface (*R*_a_) roughness are tested. It can be seen clearly that the surface roughness variation is consistent with the tested CA value. Moreover, from the SEM images of the surfaces in Table 4, the ECTFE copolymer forms numerous ridges on the membrane surface during the crystallization of the polymer and phase separation period. For such a rough surface structure and hydrophobic surface, the condensed droplets can easily detach from the membrane surface. While ensuring a sufficiently high hydrophobic membrane surface, the surface ridges can also provide the largest possible contact area between the high-humidity gas stream feed and the membrane surface, conduct a good heat exchange process, and finally separate the condensed water. More significantly, Figure 4b indicates that LEP_w_ values are all over 0.3 MPa, which is perfect since the pressure difference required for membrane condenser operation is around 0.1 MPa. It also provides strong proof that the prepared ECTFE membranes have enough potential to prevent membrane wetting from happening even during a long-term experiment. Furthermore, the LEP_w_ values consistently increased with the increasing ECTFE contents but not with the same variation of CA values. This is mainly because of the higher viscosity of dope solution resulting in a stronger squeeze among the polymer crystals during the phase-separation process. Finally, the whole structure gets denser, and the pores become smaller. From Equation (5), the LEP value is inversely proportional to membrane pore size [46,47].
(5)$${\mathrm{LEP}}_{w}=\frac{-2B\times \gamma_{L}\times \cos\theta }{r_{max}}$$
where *B* is the dimensionless geometric parameter that can be equal to 1 for assumed perfectly cylindrical pores, $\gamma_{L}$ is the liquid surface tension, θ is the CA in degrees, and $r_{max}$ (μm) is the maximum pore size.

#### 3.2.3. Mean Pore Size and Porosity of ECTFE Membrane

Table 7 shows the average pore size and porosity of the ECTFE porous membranes prepared with different polymer concentrations. As the polymer concentration increases, the average pore size gradually decreases, but with insignificant differences in the overall porosity. An inevitable result is that a denser structure forms when the polymer content is increased. The relatively denser membrane structure makes the pore space limited, the mass transferring resistance would increase in the later membrane condenser process, and finally, affect the heat transfer and condensation efficiency. Certainly, just as stated in Section 3.2.1 and shown in Equation (5), the smaller pore size can obtain a higher LEP_w_ value. The LEPw value indicates the maximum operating pressure to prevent the penetration of water. Therefore, the risk of membrane wetting for the membranes with a smaller pore size would be effectively reduced. 

#### 3.2.4. Mechanical Properties of ECTFE Membrane

The membrane condenser operation requires maintaining a certain level of pressure difference between the two sides of the membrane. Hence, the membrane is necessary to own a specific membrane strength. As shown in Figure 5, the overall variation of the tensile strength is proportional to the polymer concentration. According to the above analysis, the prepared polymer/diluent systems undergo L–L liquid phase separation, resulting in a bi-continuous structure. The SEM cross-section images indicate that the membrane structure becomes denser, and the enhanced polymer bonds with the increasing polymer content. Therefore, the tensile strength is improved. However, when the concentration increases, the space for the growth of the polymer-poor phase is limited, eventually weakening the internal connectivity and integrity between the membrane pores. The tenacity of the membrane becomes bad; as a result, elongation at break tends to decrease.

### 3.3. Dehydration by Membrane Condenser

In the membrane condenser process, the high-humidity feed stream comes into direct contact with the membrane after entering the membrane module. Due to the temperature difference between the feed stream and the membrane surface, the vapor contained in the stream condenses and forms droplets, the size of which increases gradually and finally detaches from the membrane surface. The condensation process is concurrent with mass and heat transfer simultaneously. While the dehumidified gas penetrates through the membrane to the downstream side, the condensed water on the membrane surface is withheld and collected due to the hydrophobic nature of the membrane. Moreover, a higher hydrophobic membrane surface is characterized by a higher surface roughness, which promotes the heterogeneous condensation of water molecules. Under this condition, when the droplets grow to a certain stage, they can slip off the membrane surface without affecting the subsequent contact between the membrane and the moisture. Certainly, the better hydrophobicity the membrane is, the less water left on the membrane surface; this is more beneficial for the dehumidification efficiency. Additionally, the proper pore size and porosity are also important parameters for the membrane condenser process.

The effects of different ECTFE contents on the membrane condenser efficiency were studied. First, the N_2_ flux is measured to determine the permeability, expressed in terms of gas permeance unit, GPU, of prepared ECTFE membranes (Figure 6). When the gas permeability of the tested membrane is not low, the pressure of the feed side was gradually increased during the operation period but below the LEP to avoid membrane wetting. However, a large gas permeate flux decreases the contact time between the humid gas and the membrane; thus, decreasing the final water recovery. Figure 6 indicates that the N_2_ flux decreases with increasing polymer contents. This is the expected result since the pore size and porosity decrease as the membrane structure becomes denser. As a result, the flow resistance increases, and gas flux decreases.

In this work, water and air are used to simulate a humid feed gas stream. The detailed operation parameters are shown in Table 2. The water recovery and condensate yield are shown in Figure 7a,b, respectively. The two figures show that the water recovery and condensate yield both first increase and then decrease with the rise in ECTFE content. The maximum water recovery and condensate yield values are 17.6% and 1.86 kg m^−2^ h^−1^, respectively, achieved at 30 wt% ECTFE concentration. The trend of variation of water recovery and condensate yield, as shown in Figure 7a,b, is consistent with the trend of the variation of pore size (Table 7) and contact angle (Figure 4a). The results indicated that the 30 wt% ECTFE membrane has the proper pore size and good hydrophobicity for membrane condenser application and achieved the highest water recovery and membrane condenser performance. In addition, the results indicate that the condensate yield and water recovery is proportional to the feed gas’ humidity. The ECTFE 30 wt% membrane has potential value for the water recovery process from humid flue gas in practical industrial applications. In comparison to others’ work, the overall water recovery is 20%, 35–55%, and 25% in Wang et al.’s [48]., Drioli et al.’s [34], and Macedonio et al.’s [49] research, respectively. Though the performance in this work is not the best, this is mainly related to the operating parameters and the properties of the used membrane. For example, in Drioli et al.’s work, the flow rate of the feed stream (0.076–0.38 L·min^−1^) is much lower than ours (1.5 L·min^−1^), the increasing of slow feed rate will lead to lower water recovery. Moreover, the reason why we used a high flow rate is to simulate the actual processing system and perform some preparation for our further research.

## 4. Conclusions

(1)ECTFE membranes are successfully prepared by thermally induced phase separation methods using DnOP as the diluent. The theoretical calculation of the solubility parameters indicates that the ECTFE and DnOP have good compatibility. The phase diagram of the ECTFE/DnOP binary system proves the existence of one liquid–liquid phase separation zone within the prepared polymer concentration range where a final bi-continuous structure can be obtained. Moreover, this can be clearly observed from the SEM images.(2)The surface SEM images of the ECTFE membranes present a plexiform structure with high roughness. A rough hydrophobic interface is desired as it is relatively easier for the condensed droplets to slip off the membrane surface during the membrane condenser process. The cross-section SEM images for almost all of the prepared ECTFE membranes present a bi-continuous structure. However, when the ECTFE concentration increases to over 35 wt%, the overall membrane structure become dense. As a result, the average pore size and porosity of the prepared membranes decrease.(3)The maximum contact angle of nearly 140° is obtained for membranes with an ECTFE content of 30 wt%. The LEP_w_ value of this membrane is 0.71 MPa, which is higher than the required operating pressure of the membrane condenser.(4)The membrane with the ECTFE content of 30 wt% showed the best performance in the membrane condenser process, with a water recovery of 17.6% and a condensate yield of 1.86 kg m^−2^ h^−1^.

## Figures and Tables

**Figure 1 membranes-12-00065-f001:**
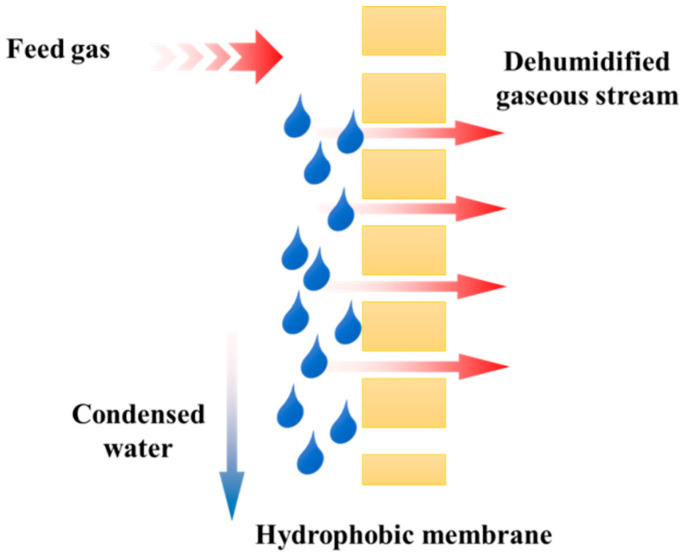
Principle of membrane condenser process.

**Figure 2 membranes-12-00065-f002:**
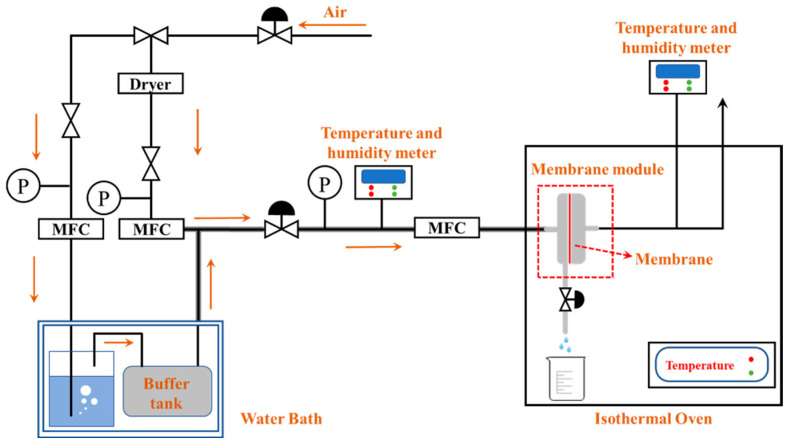
Membrane condenser experimental equipment.

**Figure 3 membranes-12-00065-f003:**
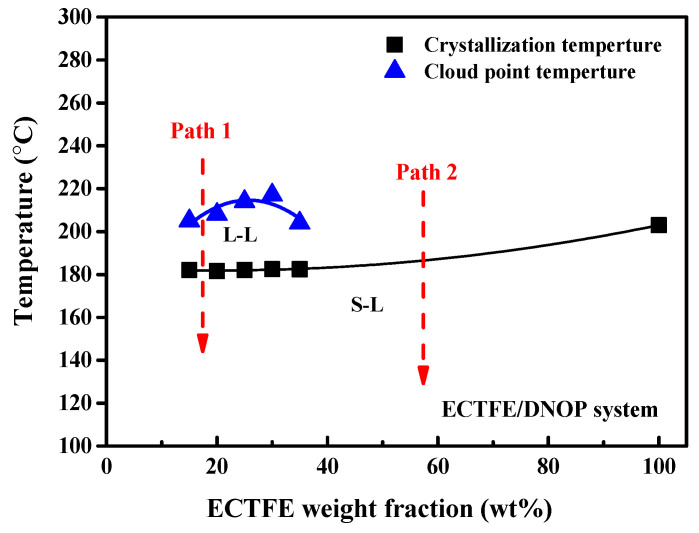
Phase diagram of ECTFE/DnOP system.

**Figure 4 membranes-12-00065-f004:**
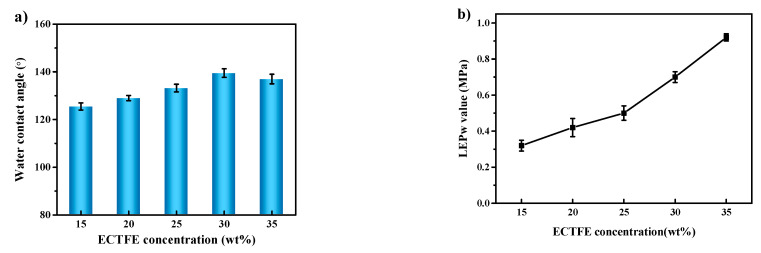
Effect of ECTFE concentrations on water contact angle (**a**) and liquid entry pressure (**b**) of the ECTFE membranes.

**Figure 5 membranes-12-00065-f005:**
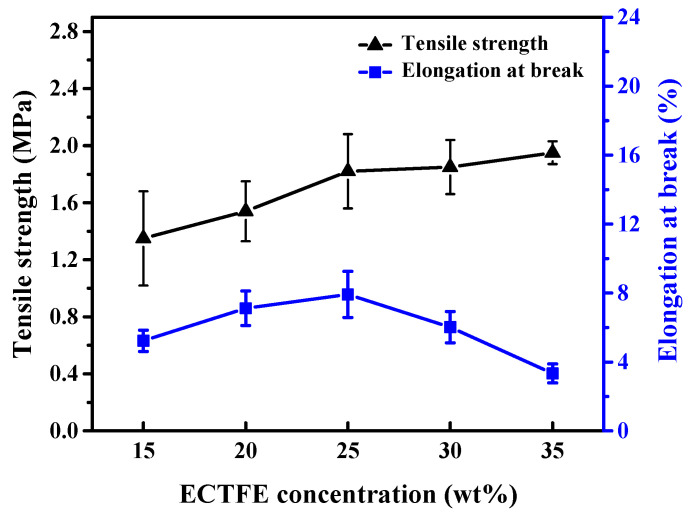
Mechanical strength of the ECTFE porous membrane prepared with different polymer concentrations.

**Figure 6 membranes-12-00065-f006:**
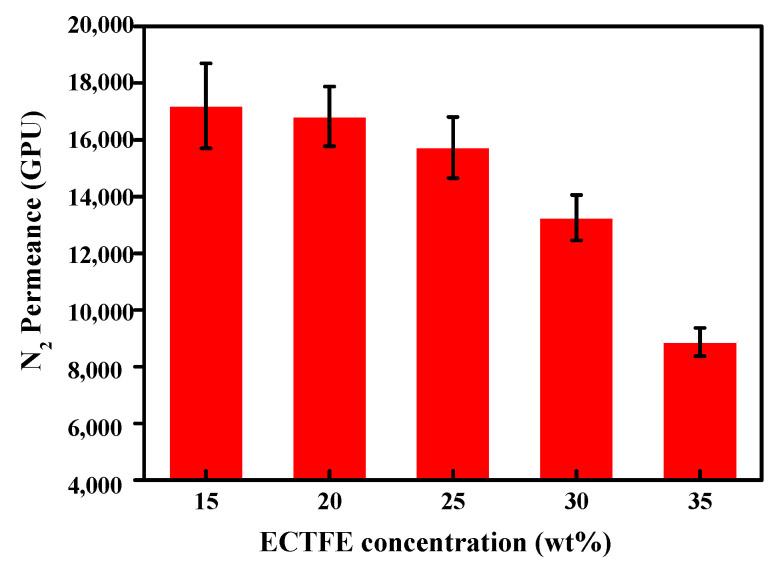
Gas flux of ECTFE membranes prepared with different polymer concentrations.

**Figure 7 membranes-12-00065-f007:**
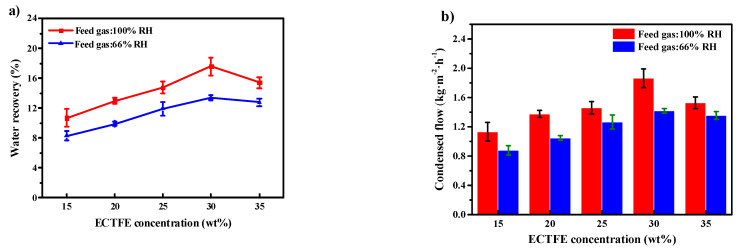
(**a**) Water recovery and (**b**) condensate yield of ECTFE membranes prepared with different polymer concentrations.

**Table 1 membranes-12-00065-t001:** The related published work about ECTFE membranes.

ECTFE Code	Solvent (s)	Membrane Type	Ref.
Halar^®^ 901	DBP, DOP, TCB	Flat sheet	[31]
Halar^®^ 901	DBP	Flat sheet	[32]
Halar^®^ 901	NMP	Flat sheet	[33]
Halar^®^ 901	GTA	Flat sheet	[34]
Halar^®^ 901	GTA/DEP	Hollow fiber	[35]
Halar^®^ 901	NMP	Hollow fiber	[36]
Halar^®^ 901	DEHA/DEP	Hollow fiber	[19]
Halar^®^ 902	DOA	Flat sheet	[37]
Halar^®^ 902	DBS/TPP	Hollow fiber	[38]
Halar^®^ 902	DEHA/DEP	Flat sheet	[18]
Halar^®^ 902	ATBC	Flat sheet	[16]
Halar^®^ 902	DEP	Flat sheet	[39]
Halar^®^ 902	DEHA/DEP	Flat sheet	[40]
Halar^®^ 902	TOTM	Flat sheet	[17]
Halar^®^ 901 and LMP ECTFE	DEA	Flat sheet	[41]

**Table 2 membranes-12-00065-t002:** The basic properties of ECTFE and DnOP.

ECTFE		DnOP	
Molecular structure: 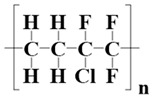		Molecular structure: 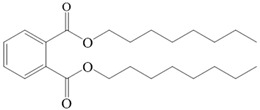	
Density (g/cm^3^)	1.68	Flash point (°C)	218
Melting point (°C)	242	Spoiling point (°C)	340

**Table 3 membranes-12-00065-t003:** Experimental parameters of membrane condenser tests.

Parameters	Value
The effective membrane area (cm^2^)	12.56
Feed gas temperature (°C)	55
Feed gas relative humidity (%RH)	100
Feed gas flow rate (L·min^−1^)	1.5
Pressure difference (kPa)	10
Temperature difference between gas and membrane surface (°C)	35

**Table 4 membranes-12-00065-t004:** Solubility parameters of ECTFE and diluents [17].

	$\delta_{d}$ (MPa^1/2^)	$\delta_{p}$ (MPa^1/2^)	$\delta_{h}$ (MPa^1/2^)	R (MPa^1/2^)
ECTFE	19.5	7.3	1.7	-
DBP	17.8	8.6	4.1	4.36
DEP	17.6	9.6	4.5	5.25
GTA	16.5	4.5	9.1	9.93
ATBC	16.02	9.1	8.55	10.86
TOTM	16.66	8.55	6.03	8.54
DnOP	16.6	6.03	3.1	5.97

**Table 5 membranes-12-00065-t005:** SEM and AFM images of ECTFE membrane.

ECTFE Content in Membranes	SEM-Surface	SEM-Cross-Section	3D AFM Image Size: 5μm × 5μm
15 wt% (thickness: 0.253 ± 0.011 mm)	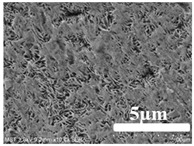	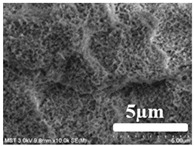	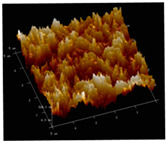
20 wt% (thickness: 0.271 ± 0.021 mm)	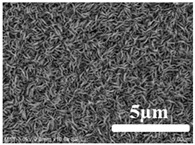	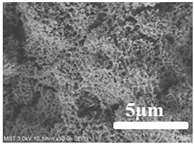	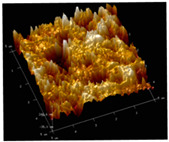
25 wt% (thickness: 0.287 ± 0.019 mm)	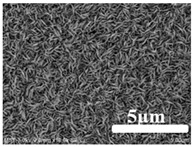	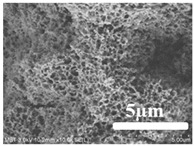	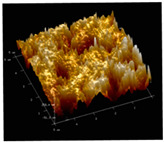
30 wt% (thickness: 0.308 ± 0.015 mm)	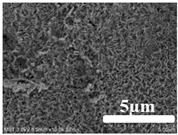	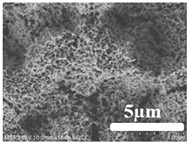	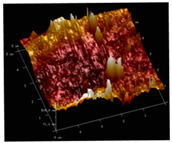
35 wt% (thickness: 0.324 ± 0.026 mm)	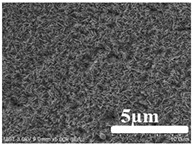	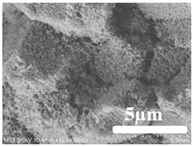	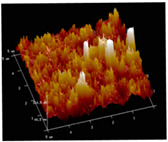

**Table 6 membranes-12-00065-t006:** Surface roughness parameters of ECTFE porous membranes prepared with different polymer concentrations.

ECTFE Content in Membranes	*R*_max_ (nm)	*R*_q_ (nm)	*R*_a_ (nm)
15 wt%	836	67.7	48.37
20 wt%	621	79.9	61.6
25 wt%	765	87.7	67.5
30 wt%	1193	121	92.4
35 wt%	812	95.5	77

**Table 7 membranes-12-00065-t007:** Pore size and porosity of prepared ECTFE membranes.

ECTFE Content in Membranes	Mean Pore Size (μm)	Porosity (%)
15 wt%	0.105 ± 0.004	60.8 ± 1.2
20 wt%	0.105 ± 0.011	59.8 ± 1.0
25 wt%	0.099 ± 0.007	61.1 ± 0.5
30 wt%	0.097 ± 0.005	60.2 ± 1.5
35 wt%	0.082 ± 0.008	56.8 ± 0.5

## Data Availability

Not applicable.

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
