# Peer review of "Preparation of ECTFE Porous Membrane for Dehumidification of Gaseous Streams through Membrane Condenser"

_membranes, 2022, doi:10.3390/membranes12010065_

Round 1

Reviewer 1 Report

1- Please rephrase all the future tense to present tense such as “will increase”.

2- Please rephrase “by Prof. Drioli’s group in 2013” to “Drioli et al. (2013)”.

3- The significance of the work should be highlighted after the aim of the present work in the last paragraph of the introduction.

4- Please use technical writing in “is kindly supplied by …”. I would suggest to remove “kindly” and write acknowledgement to the institute.

5- I would recommend to move equations 3&4 with their discussions to section 2 “Materials and Methods”.

6- The authors are recommended to cite the related works after “lower than most of the related published work [xxx]”.

7- What did the authors means by “These ridges were not observed in the published work”? Please clarify the meaning of published work.

8- It is recommended to compare the outcome of the present work with the up to date literature.

Reviewer 2 Report

The porous hydrophobic ethylene-trifluoroethylene (ECTFE) membranes were prepared by thermally induced phase separation for membrane condenser application.  The prepared ECTFE membranes were tested using a home-made membrane condenser setup. The membrane with ECTFE content of 30 wt% showed the best performance in membrane condenser process with water recovery of 17.6 % and the condensate yield of 1.86 kg‧m-2‧h-1. It is shown that the higher hydrophobic membrane surface is characterized with a higher surface roughness, which promotes  heterogeneous condensation of water molecules.

The material is quite simple, but it can be interesting. It can be published after some minor changes.

  1. It is not worth describing the idea of a membrane condenser in an abstract, since it is not presented for the first time in this article.
  2. The phrase is not quite correct«Indeed, some other non-fluorinated polymers or novel materials, like polypropylene (PP), graphene oxide (GO), carbon nanotubes (CNTs), are also used to prepare hydrophobic membranes». Not all of these substances are called polymers.Rather, they should be called materials.
  3. 12 references to the works of authors out of a total of 41 make up about 30%. This is a lot. It is desirable to decrease self-citation.

Reviewer 3 Report

  1. Past tense for methodology
  2. Please indicate how long is the sample collection time.
  3. Suggest to add mass balance calculation in the supporting information.
  4. Write unit for Equation (1), (2)-line 200
  5. Cite the published work-line 218
  6. Write the value-line 230 to line 231
  7. Check grammar - line 237
  8. Cite the work - line 262
  9. Write the value- line 282
  10. Write the unit-line 307
  11. Please further explain Table 7, try to relate the pore size of membrane in preventing the penetration of water
  12. Please further explain line 336 to 338: limited space not enhancing the polymer bonding?
  13. Suggest to include the design of the membrane module 
  14. Please justify how do you confirm that the condensation of water is happened on the membrane surface and not other parts of membrane module?
  15. Line 375: please compare the result by using non-porous disc
  16. Please mention the membrane thickness
  17. Please include membrane module in Figure 2
